# Edgeformers: Graph-Empowered Transformers for Representation Learning on Textual-Edge Networks

**Bowen Jin, Yu Zhang, Yu Meng, Jiawei Han**
Department of Computer Science, University of Illinois at Urbana-Champaign
`{bowenj4,yuz9,yumeng5,hanj}@illinois.edu`

## Abstract

Edges in many real-world social/information networks are associated with rich text information (*e.g.*, user-user communications or user-product reviews). However, mainstream network representation learning models focus on propagating and aggregating node attributes, lacking specific designs to utilize text semantics on edges. While there exist edge-aware graph neural networks, they directly initialize edge attributes as a feature vector, which cannot fully capture the contextualized text semantics of edges. In this paper, we propose Edgeformers[1], a framework built upon graph-enhanced Transformers, to perform edge and node representation learning by modeling texts on edges in a contextualized way. Specifically, in edge representation learning, we inject network information into each Transformer layer when encoding edge texts; in node representation learning, we aggregate edge representations through an attention mechanism within each node's ego-graph. On five public datasets from three different domains, Edgeformers consistently outperform state-of-the-art baselines in edge classification and link prediction, demonstrating the efficacy in learning edge and node representations, respectively.

## 1 Introduction

Networks are ubiquitous and are widely used to model interrelated data in the real world, such as user-user and user-item interactions on social media (Kwak et al., 2010; Leskovec et al., 2010) and recommender systems (Wang et al., 2019; Jin et al., 2020). In recent years, graph neural networks (GNNs) (Kipf & Welling, 2017; Hamilton et al., 2017; Velickovic et al., 2018; Xu et al., 2019) have demonstrated their power in network representation learning. However, a vast majority of GNN models leverage node attributes only and lack specific designs to capture information on edges. (We refer to these models as *node-centric* GNNs.) Yet, in many scenarios, there is rich information associated with edges in a network. For example, when a person replies to another on social media, there will be a directed edge between them accompanied by the response texts; when a user comments on an item, the user's review will be naturally associated with the user-item edge.

To utilize edge information during network representation learning, some edge-aware GNNs (Gong & Cheng, 2019; Jiang et al., 2019; Yang & Li, 2020; Jo et al., 2021) have been proposed. Nevertheless, these studies assume the information carried by edges can be directly described as an attribute vector. This assumption holds well when edge features are categorical (*e.g.*, bond features in molecular graphs (Hu et al., 2020) and relation features in knowledge graphs (Schlichtkrull et al., 2018)). However, effectively modeling free-text edge information in edge-aware GNNs has remained elusive, mainly because bag-of-words and context-free embeddings (Mikolov et al., 2013) used in previous edge-aware GNNs cannot fully capture contextualized text semantics. For example, "*Byzantine*" in history book reviews and "*Byzantine*" in distributed system papers should have different meanings given their context, but they correspond to the same entry in a bag-of-words vector and have the same context-free embedding.

To accurately capture contextualized semantics, a straightforward idea is to integrate pretrained language models (PLMs) (Devlin et al., 2019; Liu et al., 2019; Clark et al., 2020) with GNNs. In node-centric GNN studies, this idea has been instantiated by a PLM-GNN cascaded architecture (Fang et al., 2020; Li et al., 2021; Zhu et al., 2021), where text information is first encoded by a PLM and then aggregated by a GNN. However, such architectures process text and graph signals one after

---

[1]Code can be found at `https://github.com/PeterGriffinJin/Edgeformers`.

the other, and fail to simultaneously model the deep interactions between both types of information. This could be a loss to the text encoder because network signals are often strong indicators to text semantics. For example, a brief political tweet may become more comprehensible if the stands of the two communicators are known. To deeply couple PLMs and GNNs, the recent GraphFormers model (Yang et al., 2021) proposes a GNN-nested PLM architecture to inject network information into the text encoding process. They introduce GNNs nested in between Transformer layers so that the center node encoding not only leverages its own textual information, but also aggregates the signals from its neighbors. Nevertheless, they assume that only nodes are associated with textual information and cannot be easily adapted to handle text-rich edges.

To effectively model the textual and network structure information via a unified encoder architecture, in this paper, we propose a novel network representation learning framework, Edgeformers, that leverage graph-enhanced Transformers to model edge texts in a contextualized way. Edgeformers include two architectures, Edgeformer-E and Edgeformer-N, for edge and node representation learning, respectively. In Edgeformer-E, we add virtual node tokens to each Transformer layer inside the PLM when encoding edge texts. Such an architecture goes beyond the PLM-GNN cascaded architecture and enables deep, layer-wise interactions between network and text signals to produce edge representations. In Edgeformer-N, we aggregate the network-and-text-aware edge representations to obtain node representations through an attention mechanism within each node's ego-graph. The two architectures can be trained via edge classification (which relies on good edge representations) and link prediction (which relies on good node representations) tasks, respectively. To summarize, our main contributions are as follows:

- Conceptually, we identify the importance of modeling text information on network edges and formulate the problem of representation learning on textual-edge networks.

- Methodologically, we propose Edgeformers (*i.e.*, Edgeformer-E and Edgeformer-N), two graph-enhanced Transformer architectures, to deeply couple network and text information in a contextualized way for edge and node representation learning.

- Empirically, we conduct experiments on five public datasets from different domains and demonstrate the superiority of Edgeformers over various baselines, including node-centric GNNs, edge-aware GNNs, and PLM-GNN cascaded architectures.

## 2 PRELIMINARIES

### 2.1 TEXTUAL-EDGE NETWORKS

In a textual-edge network, each edge is associated with texts. We view the texts on each edge as a document, and all such documents constitute a corpus $\mathcal{D}$. Since the major goal of this work is to explore the effect of textual information on edges, we assume there is no auxiliary information (*e.g.*, categorical or textual attributes) associated with nodes in the network.

**Definition 1** *(Textual-Edge Networks) A textual-edge network is defined as $\mathcal{G} = (\mathcal{V}, \mathcal{E}, \mathcal{D})$, where $\mathcal{V}$, $\mathcal{E}$, $\mathcal{D}$ represent the sets of nodes, edges, and documents, respectively. Each edge $e_{ij} \in \mathcal{E}$ is associated with a document $d_{ij} \in \mathcal{D}$.*

To give an example of textual-edge networks, consider a review network (*e.g.*, Amazon (He & McAuley, 2016)) where nodes are users and items. If a user $v_i$ writes a review about an item $v_j$, there will be an edge $e_{ij}$ connecting them, and the review text will be the associated document $d_{ij}$.

### 2.2 TRANSFORMER

Many PLMs (*e.g.*, BERT (Devlin et al., 2019)) adopt a multi-layer Transformer architecture (Vaswani et al., 2017) to encode texts. Each Transformer layer utilizes a multi-head self-attention mechanism to obtain a contextualized representation of each text token. Specifically, let $\boldsymbol{H}^{(l)} = [\boldsymbol{h}_1^{(l)}, \boldsymbol{h}_2^{(l)}, ..., \boldsymbol{h}_n^{(l)}]$ denote the output sequence of the $l$-th Transformer layer, where $\boldsymbol{h}_i^{(l)} \in \mathcal{R}^d$ is the hidden representation of the text token at position $i$. Then, in the $(l + 1)$-th Transformer layer, the multi-head self-attention (MHA) is calculated as

$$\text{MHA}(\boldsymbol{H}^{(l)}) = \overset{k}{\underset{t=1}{\|}} \text{head}^t(\boldsymbol{H}_t^{(l)}) \tag{1}$$

$$\text{head}^t(\boldsymbol{H}_t^{(l)}) = \boldsymbol{V}_t^{(l)} \cdot \text{softmax}(\frac{\boldsymbol{K}_t^{(l)\top} \boldsymbol{Q}_t^{(l)}}{\sqrt{d/k}}) \tag{2}$$

$$\boldsymbol{Q}_t^{(l)} = \boldsymbol{W}_{Q,t}^{(l)} \boldsymbol{H}_t^{(l)}, \quad \boldsymbol{K}_t^{(l)} = \boldsymbol{W}_{K,t}^{(l)} \boldsymbol{H}_t^{(l)}, \quad \boldsymbol{V}_t^{(l)} = \boldsymbol{W}_{V,t}^{(l)} \boldsymbol{H}_t^{(l)}, \tag{3}$$

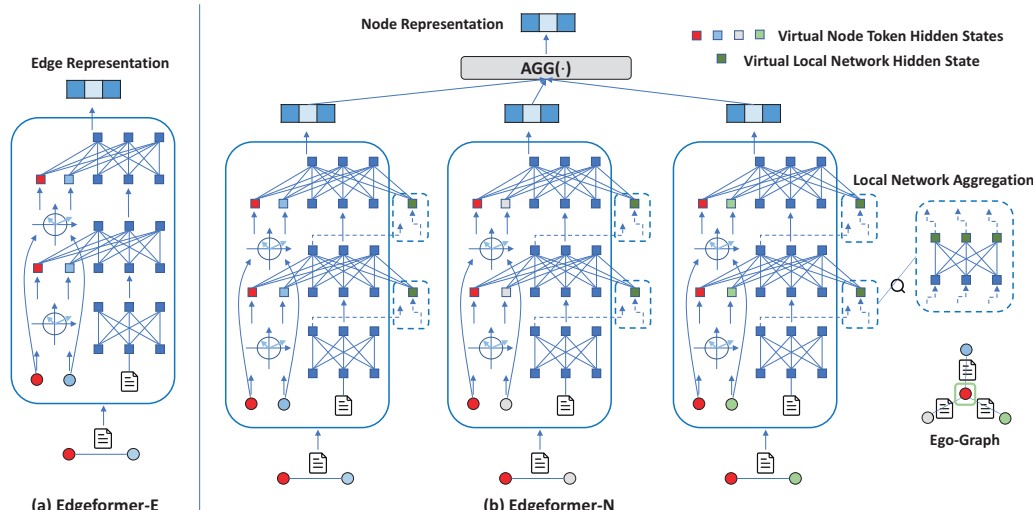

Figure 1: Model Framework Overview. (a) An illustration of Edgeformer-E for edge representation learning, where virtual node token hidden states are concatenated to the edge text original token hidden states to inject network signal into edge text encoding. (b) An illustration of Edgeformer-N for node representation learning, where Edgeformer-E is enhanced by local network structure virtual token hidden state and edge representations are aggregated to obtain node representation.

where $\boldsymbol{W}_{Q,t}, \boldsymbol{W}_{K,t}, \boldsymbol{W}_{V,t}$ are query, key, and value matrices to be learned by the model, $k$ is the number of attention head and $\|$ is the concatenate operation.

## 2.3 PROBLEM DEFINITIONS

Our general goal is to learn meaningful edge and node embeddings in textual-edge networks so as to benefit downstream tasks. To be specific, we consider the following two tasks focusing on edge representation learning and node representation learning, respectively.

The first task is *edge classification*, which relies on learning a good representation $\boldsymbol{h}_e$ of an edge $e \in \mathcal{E}$. We assume each edge $e_{ij}$ belongs to a category $y \in \mathcal{Y}$. The category can be indicated by its associated text $d_{ij}$ and/or the nodes $v_i$ and $v_j$. For example, in the Amazon review network, $\mathcal{Y} = \{1\text{-star, 2-star, ..., 5-star}\}$. The category of $e_{ij}$ reflects how the user $v_i$ is satisfied with the item $v_j$, which may be expressed by the sentiment of $d_{ij}$ and/or implied by $v_i$'s preference and $v_j$'s quality. Given a review, the task is to predict its category based on review text and user/item information.

**Definition 2** *(Edge Classification) In a textual-edge network $\mathcal{G} = (\mathcal{V}, \mathcal{E}, \mathcal{D})$, we can observe the category of some edges $\mathcal{E}_{train} \subseteq \mathcal{E}$. Given an edge $e_{ij} \in \mathcal{E} \backslash \mathcal{E}_{train}$, predict its category $y \in \mathcal{Y}$ based on $d_{ij} \in \mathcal{D}$ and $v_i, v_j \in \mathcal{V}$.*

The second task is *link prediction*, which relies on learning an accurate representation $\boldsymbol{h}_{v_i}$ of a node $v_i \in \mathcal{V}$. Given two nodes $v_i$ and $v_j$, the task is to predict whether there is an edge between them. Note that, unlike edge classification, we no longer have the text information $d_{ij}$ (because we even do not know whether $e_{ij}$ exists). Instead, we need to exploit other edges (local network structure) involving $v_i$ or $v_j$ as well as their text to learn node representations $\boldsymbol{h}_{v_i}$ and $\boldsymbol{h}_{v_j}$. For example, in the Amazon review network, we aim to predict whether a user will be satisfied with a product according to the user's reviews towards other products and the item's reviews from other users.

**Definition 3** *(Link Prediction) In a textual-edge network $\mathcal{G} = (\mathcal{V}, \mathcal{E}, \mathcal{D})$, we can observe some edges $\mathcal{E}_{train} \subseteq \mathcal{E}$ and their associated text. Given $v_i, v_j \in \mathcal{V}$ where $e_{ij} \notin \mathcal{E}_{train}$, predict whether $e_{ij} \in \mathcal{E}$.*

## 3 PROPOSED METHOD

In this section, we present our Edgeformers framework. Based on the two tasks mentioned in Section 2.3, we first introduce how we conduct *edge* representation learning by jointly considering text and network information via a Transformer-based architecture (Edgeformer-E). Then, we illustrate how to perform *node* representation learning using the edge representation learning module as building blocks (Edgeformer-N). The overview of Edgeformers is shown in Figure 1.

## 3.1 EDGE REPRESENTATION LEARNING (EDGEFORMER-E)

**Network-aware Edge Text Encoding with Virtual Node Tokens.** Encoding $d_{ij}$ in a textual-edge network is different from encoding plain text, mainly because edge texts are naturally accompanied by network structure information, which can provide auxiliary signals. Given that text semantics can be well captured by a multi-layer Transformer architecture (Devlin et al., 2019), we propose a simple and effective way to inject network signals into the Transformer encoding process. The key idea is to introduce *virtual node tokens*. Given an edge $e_{ij} = (v_i, v_j)$ and its associated texts $d_{ij}$, let $\boldsymbol{H}_{e_{ij}}^{(l)} \in \mathcal{R}^{d \times n}$ denote the output representations of all text tokens in $d_{ij}$ after the $l$-th model layer ($l \geqslant 1$). In each layer, we introduce two *virtual node tokens* to represent $v_i$ and $v_j$, respectively. Their embeddings are denoted as $\boldsymbol{z}_{v_i}^{(l)}$ and $\boldsymbol{z}_{v_j}^{(l)} \in \mathcal{R}^d$, which are concatenated to the text token sequence hidden states as follows:

$$\widetilde{\boldsymbol{H}}_{e_{ij}}^{(l)} = \boldsymbol{z}_{v_i}^{(l)} \parallel \boldsymbol{z}_{v_j}^{(l)} \parallel \boldsymbol{H}_{e_{ij}}^{(l)}. \tag{4}$$

After the concatenation, $\widetilde{\boldsymbol{H}}_{e_{ij}}^{(l)}$ contains information from both $e_{ij}$'s associated text $d_{ij}$ and its involving nodes $v_i$ and $v_j$. To let text token representations carry node signals, we adopt a multi-head attention mechanism:

$$\mathrm{MHA}(\boldsymbol{H}_{e_{ij}}^{(l)}, \widetilde{\boldsymbol{H}}_{e_{ij}}^{(l)}) = \mathop{\parallel}_{t=1}^{k} \mathrm{head}^t(\boldsymbol{H}_{e_{ij},t}^{(l)}, \widetilde{\boldsymbol{H}}_{e_{ij},t}^{(l)}), \tag{5}$$

$$\boldsymbol{Q}_t^{(l)} = \boldsymbol{W}_{Q,t}^{(l)} \boldsymbol{H}_{e_{ij},t}^{(l)}, \quad \boldsymbol{K}_t^{(l)} = \boldsymbol{W}_{K,t}^{(l)} \widetilde{\boldsymbol{H}}_{e_{ij},t}^{(l)}, \quad \boldsymbol{V}_t^{(l)} = \boldsymbol{W}_{V,t}^{(l)} \widetilde{\boldsymbol{H}}_{e_{ij},t}^{(l)}. \tag{6}$$

In Eq. (5), the multi-head attention is asymmetric (*i.e.*, the keys $\boldsymbol{K}$ and values $\boldsymbol{V}$ are augmented with virtual node embeddings but queries $\boldsymbol{Q}$ are not) to avoid network information being overwritten by text signals. This design has been used in existing studies Yang et al. (2021), and offers better effectiveness than the original self-attention mechanism according to our experiments in Section 4.2. The output of MHA includes updated node-aware representations of text tokens. Then, following the Transformer architecture (Vaswani et al., 2017), the updated representations will go through a feed-forward network (FFN) to finish our $(l+1)$-th model layer encoding. Formally,

$$\boldsymbol{H}_{e_{ij}}^{(l)'} = \mathrm{Normalize}(\boldsymbol{H}_{e_{ij}}^{(l)} + \mathrm{MHA}(\boldsymbol{H}_{e_{ij}}^{(l)}, \widetilde{\boldsymbol{H}}_{e_{ij}}^{(l)})), \tag{7}$$

$$\boldsymbol{H}_{e_{ij}}^{(l+1)} = \mathrm{Normalize}(\boldsymbol{H}_{e_{ij}}^{(l)'} + \mathrm{FFN}(\boldsymbol{H}_{e_{ij}}^{(l)'})), \tag{8}$$

where $\mathrm{Normalize}(\cdot)$ is the layer normalization function. After $L$ model layers, the final representation of the [CLS] token will be used as the edge representation of $e_{ij}$, *i.e.*, $\boldsymbol{h}_{e_{ij}} = \boldsymbol{H}_{e_{ij}}^{(L)}[\mathrm{CLS}]$.

**Representation of Virtual Node Tokens.** The virtual node representation $\boldsymbol{z}_{v_i}^{(l)}$ used in Eq.(4) is obtained by a layer-specific mapping of the initial node embedding $\boldsymbol{z}_{v_i}^{(0)}$. Formally,

$$\boldsymbol{z}_{v_i}^{(l)} = \boldsymbol{W}_n^{(l)} \boldsymbol{z}_{v_i}^{(0)}, \tag{9}$$

where $\boldsymbol{W}_n^{(l)} \in \mathcal{R}^{d \times d'}$ is the mapping matrix for the $l$-th layer. The large population of nodes will introduce a large number of parameters to our framework, which may finally lead to model underfitting. As a result, in Edgeformers, we set the initial node embedding to be low-dimensional (*e.g.*, $\boldsymbol{z}_{v_i}^{(0)} \in \mathbb{R}^{64}$) and project it to the high-dimensional token representation space (*e.g.*, $\boldsymbol{z}_{v_i}^{(l)} \in \mathbb{R}^{768}$). Note that it is possible to go beyond the linear mapping in Eq. (9) and use structure-aware encoders such as GNNs to obtain $\boldsymbol{z}_{v_i}^{(l)}$, and we leave such extensions for future studies.

## 3.2 TEXT-AWARE NODE REPRESENTATION LEARNING (EDGEFORMER-N)

In this section, we first discuss how to perform text-aware node representation learning by taking the aforementioned edge representation learning module (*i.e.*, Edgeformer-E) as building blocks. Then, we propose to enhance the edge representation learning module with the target node's additional local network structure.

**Aggregating Edge Representations.** Since the edge representations learned by Edgeformer-E capture both text semantics and network structure information, a straightforward way to obtain a node representation is to aggregate the representations of all edges involving the node. Given a node $v_i$, its representation $\boldsymbol{h}_{v_i}$ is given by

$$\boldsymbol{h}_{v_i} = \mathrm{AGG}(\{\boldsymbol{h}_{e_{ij}} | e_{ij} \in \mathcal{N}_e(v_i)\}), \tag{10}$$

where $\mathcal{N}_e(v_i)$ is the set of edges containing $v_i$. AGG($\cdot$) can be any permutation invariant function such as mean($\cdot$) or max($\cdot$). Here, we instantiate AGG($\cdot$) with an attention-based aggregation:

$$\alpha_{e_{ij},v_i} = \text{softmax}(\boldsymbol{h}_{e_{ij}}^\top \boldsymbol{W}_s \boldsymbol{z}_{v_i}^{(0)}), \quad \boldsymbol{h}_{v_i} = \sum_{e_{ij} \in \mathcal{N}_e(v_i)} \alpha_{e_{ij},v_i} \boldsymbol{h}_{e_{ij}}, \tag{11}$$

where $\boldsymbol{W}_s \in \mathcal{R}^{d \times d'}$ is a learnable scoring matrix.

**Enhancing Edge Representations with the Node's Local Network Structure.** Since we are aggregating information from multiple edges, it is intuitive that they can mutually improve each other's representation by providing auxiliary semantic signals. For example, given a conversation about "Transformers" and their participants' other conversations centered around "machine learning", it is more likely that the term "Transformers" refers to a deep learning architecture rather than a character in the movie. To implement this intuition in the edge representation learning module, we introduce the third virtual token hidden state $\bar{\boldsymbol{h}}_{e_{ij}|v_i}^{(l)}$ during edge encoding:

$$\widetilde{\boldsymbol{H}}_{e_{ij}|v_i}^{(l)} = \boldsymbol{z}_{v_i}^{(l)} \,\|\, \boldsymbol{z}_{v_j}^{(l)} \,\|\, \bar{\boldsymbol{h}}_{e_{ij}|v_i}^{(l)} \,\|\, \boldsymbol{H}_{e_{ij}}^{(l)}, \tag{12}$$

where $\bar{\boldsymbol{h}}_{e_{ij}|v_i}^{(l)}$ is the contextualized representation of $e_{ij}$ given target node $v_i$'s local network structure. Now we introduce how to calculate $\bar{\boldsymbol{h}}_{e_{ij}|v_i}^{(l)}$ by aggregating information from $\mathcal{N}_e(v_i)$.

**Representation of $\bar{\boldsymbol{h}}_{e_{ij}|v_i}^{(l)}$.** For each edge $e_{is} \in \mathcal{N}_e(v_i)$ (including $e_{ij}$), we treat the hidden state of its [CLS] token after the $l$-th layer as its representation (*i.e.*, $\boldsymbol{h}_{e_{is}}^{(l)} = \boldsymbol{H}_{e_{is}}^{(l)}[\text{CLS}]$). To obtain $\bar{\boldsymbol{h}}_{e_{is}|v_i}^{(l)}$, we adopt MHA to let all edges in $\mathcal{N}_e(v_i)$ interact with each other.

$$\left[ ..., \bar{\boldsymbol{h}}_{e_{ij}|v_i}^{(l)}, ..., \bar{\boldsymbol{h}}_{e_{is}|v_i}^{(l)}, ... \right] = \text{MHA} \left( \left[ ..., \boldsymbol{h}_{e_{ij}}^{(l)}, ..., \boldsymbol{h}_{e_{is},...}^{(l)} \right] \right). \tag{13}$$

In the equation above, $\left[ ..., \boldsymbol{h}_{e_{ij}}^{(l)}, ..., \boldsymbol{h}_{e_{is},...}^{(l)} \right]$ contains the $l$-th layer representations of all edges involving $v_i$. Therefore, after MHA, the edge representation $\bar{\boldsymbol{h}}_{e_{ij}|v_i}^{(l)}$ essentially aggregates information from $v_i$'s local network structure $\mathcal{N}_e(v_i)$.

**Connection between Edgeformer-N and GNNs.** According to Figure 1, Edgeformer-N adopts a Transformer-based architecture. Meanwhile, it can also be viewed as a GNN model. Indeed, GNN models (Wu et al., 2020; Yang et al., 2020) mainly adopt a propagation-aggregation paradigm to obtain node representations:

$$\boldsymbol{a}_{ij}^{(l-1)} = \text{PROP}^{(l)} \left( \boldsymbol{h}_i^{(l-1)}, \boldsymbol{h}_j^{(l-1)} \right), \left( \forall j \in \mathcal{N}(i) \right); \ \boldsymbol{h}_i^{(l)} = \text{AGG}^{(l)} \left( \boldsymbol{h}_i^{(l-1)}, \{ \boldsymbol{a}_{ij}^{(l-1)} | j \in \mathcal{N}(i) \} \right). \tag{14}$$

Analogously, in Edgeformer-N, Eq. (13) can be treated as the propagation function $\text{PROP}^{(l)}$, and the aggregation step $\text{AGG}^{(l)}$ is the combination of Eqs. (12), (7), (8), and (10).

## 3.3 TRAINING

As mentioned in Section 2.3, we consider edge classification and link prediction as two tasks to train Edgeformer-E and Edgeformer-N, respectively.

**Edge Classification.** For Edgeformer-E (*i.e.*, edge representation learning), we adopt supervised training, the objective function of which is as follows.

$$\mathcal{L}_e = -\sum_{e_{ij}} \boldsymbol{y}_{e_{ij}}^\top \log \hat{\boldsymbol{y}}_{e_{ij}} + (1 - \boldsymbol{y}_{e_{ij}})^\top \log(1 - \hat{\boldsymbol{y}}_{e_{ij}}), \tag{15}$$

where $\hat{\boldsymbol{y}}_{e_{ij}} = f(\boldsymbol{h}_{e_{ij}})$ is the predicted category distribution of $e_{ij}$ and $f(\cdot)$ is a learnable classifier.

**Link Prediction.** For Edgeformer-N (*i.e.*, node representation learning), we conduct unsupervised training, where the objective function is as follows.

$$\mathcal{L}_n = \sum_{v \in \mathcal{V}} \sum_{u \in \mathcal{N}_n(v)} -\log \frac{\exp(\boldsymbol{h}_v^\top \boldsymbol{h}_u)}{\exp(\boldsymbol{h}_v^\top \boldsymbol{h}_u) + \sum_{u'} \exp(\boldsymbol{h}_v^\top \boldsymbol{h}_{u'})}. \tag{16}$$

Here, $\mathcal{N}_n(v)$ is the set of $v$'s node neighbors and $u'$ denotes a random negative sample. In our implementation, we utilize "in-batch negative samples" (Karpukhin et al., 2020) to reduce encoding and training costs.

Table 1: Edge classification performance on Amazon-Movie, Amazon-App, Goodreads-Crime, and Goodreads-Children.

| Model | Amazon-Movie | | Amazon-Apps | | Goodreads-Crime | | Goodreads-Children | |
|---|---|---|---|---|---|---|---|---|
| | Macro-F1 | Micro-F1 | Macro-F1 | Micro-F1 | Macro-F1 | Micro-F1 | Macro-F1 | Micro-F1 |
| TF-IDF | 50.01 | 64.22 | 48.30 | 62.88 | 43.07 | 51.72 | 39.42 | 49.90 |
| TF-IDF+nodes | 53.59 | 66.34 | 50.56 | 65.08 | 49.35 | 57.50 | 47.32 | 56.78 |
| BERT | 61.38 | 71.36 | 59.11 | 69.27 | 56.41 | 61.29 | 51.57 | 57.72 |
| BERT+nodes | *63.00* | *72.45* | *59.72* | *70.82* | *58.64* | *65.02* | *54.42* | *60.46* |
| Edgeformer-E | **64.18** | **73.59** | **60.67** | **71.28** | **61.03** | **65.86** | **57.45** | **61.71** |

**Overall Algorithm.** The workflow of our edge representation learning (Edgeformer-E) and node representation learning (Edgeformer-N) algorithms can be found in Alg. 1 and Alg. 2, respectively.

**Complexity Analysis.** Given a node involved in $N$ edges, and each edge has $P$ text tokens, the time complexity of *edge* encoding for each Edgeformer-E layer is $\mathcal{O}(P^2)$ (the same as one vanilla Transformer layer). The time complexity of *node* encoding for each Edgeformer-N layer is $\mathcal{O}(NP^2 + N^2)$. For most nodes in the network, we can assume $N^2 \ll NP^2$, so the complexity is roughly $\mathcal{O}(NP^2)$ (the same as one PLM-GNN cascaded layer). For more discussions about time complexity, please refer to Section 4.5.

## 4 EXPERIMENTS

In this section, we first introduce five datasets. Then, we demonstrate the effectiveness of Edgeformers on both edge-level (*e.g.*, edge classification) and node-level (*e.g.*, link prediction) tasks. Finally, we conduct visualization and efficiency analysis to further understand Edgeformers.

### 4.1 DATASETS

We run experiments on three real-world networks: Amazon (He & McAuley, 2016), Goodreads (Wan et al., 2019), and StackOverflow[2]. Amazon is a user-item interaction network, where reviews are treated as text on edges; Goodreads is a reader-book network, where readers' comments are used as edge text information; StackOverflow is an expert-question network, and there will an edge when an expert posts an answer to a question. Since Amazon and Goodreads both have multiple domains, we select two domains for each of them. In total, there are five datasets used in evaluation (*i.e.*, Amazon-Movie, Amazon-Apps, Goodreads-Crime, Goodreads-Children, StackOverflow). Dataset statistics can be found in Appendix A.1.

### 4.2 TASK FOR EDGE REPRESENTATION LEARNING

**Baselines.** We compare our Edgeformer-E model with a bag-of-words method (TF-IDF (Robertson & Walker, 1994)) and a pretrained language model (BERT (Devlin et al., 2019)). Both baselines are further enhanced with network information by concatenating the node embedding $z_i$ with the bag-of-words vector (TF-IDF+nodes) or appending it to the input token sequence (BERT+nodes).

**Edge Classification.** The model is asked to predict the category of each edge based on its associated text and local network structure. There are 5 categories for edges in Amazon (*i.e.*, 1-star, ..., 5-star) and 6 categories for edges in Goodreads (*i.e.*, 0-star, ..., 5-star).

For TF-IDF methods, the dimension of the bag-of-words vector is 2000. BERT-involved models and Edgeformer-E have the same model size ($L = 12, d = 768$) and are initialized by the same checkpoint[3]. The dimension of initial node embeddings $d'$ is set to be 64. We use AdamW as the optimizer with $(\epsilon, \beta_1, \beta_2) = (1\text{e-}8, 0.9, 0.999)$. The learning rate is 1e-5. The early stopping patience is 3 epochs. The batch size is 25. Macro-F1 and Micro-F1 are used as evaluation metrics. For BERT-involved models, parameters in BERT are trainable.

Table 1 summarizes the performance comparison on the five datasets. From the table, we can observe that: (a) our Edgeformer-E consistently outperforms all the baseline methods; (b) PLM-based methods (*i.e.*, BERT, BERT+nodes, and Edgeformer-E) can have more promising results than bag-of-words methods (*i.e.*, TF-IDF and TF-IDF+nodes); (c) injecting node information can significantly improve performance if we compare TF-IDF with TF-IDF+nodes or compare BERT with BERT+nodes; (d) the performance of Edgeformer-E is better than that of directly appending node embeddings to the

---

[2] https://www.kaggle.com/datasets/stackoverflow/stackoverflow
[3] https://huggingface.co/bert-base-uncased

Table 2: Link prediction performance (on the testing set) on Amazon-Movie, Amazon-Apps, Goodreads-Crime, Goodreads-Children, and StackOverflow. $\Delta$ denotes the relative improvement of our model comparing with the best baseline.

| Model | Amazon-Movie | | Amazon-Apps | | Goodreads-Crime | | Goodreads-Children | | StackOverflow | |
|---|---|---|---|---|---|---|---|---|---|---|
| | MRR | NDCG | MRR | NDCG | MRR | NDCG | MRR | NDCG | MRR | NDCG |
| MF | 0.2032 | 0.3546 | 0.1482 | 0.3052 | 0.1923 | 0.3443 | 0.1137 | 0.2716 | 0.1040 | 0.2642 |
| MeanSAGE | 0.2138 | 0.3657 | 0.1766 | 0.3343 | 0.1832 | 0.3368 | 0.1066 | 0.2647 | 0.1174 | 0.2768 |
| MaxSAGE | 0.2178 | 0.3694 | 0.1674 | 0.3258 | 0.1846 | 0.3387 | 0.1066 | 0.2647 | 0.1173 | 0.2769 |
| GIN | 0.2140 | 0.3648 | 0.1797 | 0.3362 | 0.1846 | 0.3374 | 0.1128 | 0.2700 | 0.1189 | 0.2778 |
| CensNet | 0.2048 | 0.3568 | 0.1894 | 0.3457 | 0.1880 | 0.3398 | 0.1157 | 0.2726 | 0.1235 | 0.2806 |
| NENN | 0.2565 | 0.4032 | 0.1996 | 0.3552 | 0.2173 | 0.3670 | 0.1297 | 0.2854 | 0.1257 | 0.2854 |
| BERT | 0.2391 | 0.3864 | 0.1790 | 0.3350 | 0.1986 | 0.3498 | 0.1274 | 0.2836 | 0.1666 | 0.3252 |
| BERT+MaxSAGE | 0.2780 | 0.4224 | 0.2055 | 0.3602 | 0.2193 | 0.3694 | 0.1312 | 0.2872 | 0.1681 | 0.3264 |
| BERT+MeanSAGE | 0.2491 | 0.3972 | 0.1983 | 0.3540 | 0.1952 | 0.3477 | 0.1223 | 0.2791 | 0.1678 | 0.3264 |
| BERT+GIN | 0.2573 | 0.4037 | 0.2000 | 0.3552 | 0.2007 | 0.3522 | 0.1238 | 0.2801 | *0.1708* | *0.3279* |
| GraphFormers | 0.2756 | 0.4198 | 0.2066 | 0.3607 | 0.2176 | 0.3684 | 0.1323 | 0.2887 | 0.1693 | 0.3278 |
| BERT+CensNet | 0.1919 | 0.3462 | 0.1544 | 0.3132 | 0.1437 | 0.3000 | 0.0847 | 0.2436 | 0.1173 | 0.2789 |
| BERT+NENN | *0.2821* | *0.4256* | *0.2127* | *0.3666* | *0.2262* | *0.3756* | *0.1365* | *0.2925* | 0.1619 | 0.3215 |
| Edgeformer-N | **0.2919** | **0.4344** | **0.2239** | **0.3771** | **0.2395** | **0.3875** | **0.1446** | **0.3000** | **0.1754** | **0.3339** |
| $+ \Delta \%$ | **3.5%** | **2.1%** | **5.3%** | **2.9%** | **5.9%** | **3.2%** | **5.9%** | **2.6%** | **2.7%** | **1.8%** |

input token sequence (*i.e.*, BERT+nodes), possibly because network information is overwritten by text signals in BERT+nodes' deeper layers.

### 4.3 TASKS FOR NODE REPRESENTATION LEARNING

**Baselines.** We compare Edgeformer-N with several **vanilla GNN** models and **PLM-integrated GNN** models. **Vanilla GNN** models include *node-centric GNNs* such as MeanSAGE (Hamilton et al., 2017), MaxSAGE (Hamilton et al., 2017) and GIN (Xu et al., 2019), and *edge-aware GNNs* such as CensNet (Jiang et al., 2019) and NENN (Yang & Li, 2020). All **vanilla** *edge-aware GNNs* models use bag-of-words as initial edge feature representations. **PLM-integrated GNN** models utilize a PLM (Devlin et al., 2019) to obtain text representations on edges and adopt a GNN to obtain node representations by aggregating edge representations. Baselines include BERT+MeanSAGE (Hamilton et al., 2017), BERT+MaxSAGE (Hamilton et al., 2017), BERT+GIN (Xu et al., 2019), BERT+CensNet (Jiang et al., 2019), BERT+NENN (Yang & Li, 2020), and GraphFormers (Yang et al., 2021). To verify the importance of both text and network information in text-rich networks, we also include matrix factorization (MF) (Qiu et al., 2018) and vanilla BERT (Devlin et al., 2019) in the comparison.

**Link Prediction.** The task is to predict whether there will be an edge between two target nodes, given their local network structures. Specifically, in the Amazon and Goodreads datasets, given the target user's reviews to other items/books and the target item/book's reviews from other users, we aim to predict whether there will be a 5-star link between the target user and the target item/book. In the StackOverflow dataset, we aim to predict whether the target expert can give an answer to the target question. We use MRR and NDCG as evaluation metrics.

For vanilla GNN models, we find that adopting MF node embeddings as initial node embeddings can help them obtain better performance (Lv et al., 2021). For edge-aware GNNs, bag-of-words vectors are used as edge features, the size of which is set as 2000. For BERT-involved models, the training parameters are the same as 4.2. During the testing stage, all methods are evaluated with samples in the batch for efficiency, *i.e.*, each query node is provided with one positive key node and 99 randomly sampled negative key nodes. More details can be found in Appendix A.8.

Table 2 shows the performance comparison. From the table, we can find that: (a) Edgeformer-N outperforms all the baseline methods consistently; (b) BERT-based methods can have significantly better performance than bag-of-words GNN methods, which demonstrates the importance of contextualized text semantics encoding; (c) edge-aware methods can have better performance, but it depends on how the edge information contributes to node representation learning; (d) our Edgeformer-N is the best since it takes edge text into consideration and deeply integrates text encoding and local network structure encoding.

**Ablation Study.** We further conduct an ablation study to validate the effectiveness of all the three virtual tokens on node representation learning. The three virtual token hidden states are deleted respectively from the whole model and the results are shown in Table 3. From the table, we can find that Edgeformer-N generally outperforms all the model variants on all the datasets, except for that

Table 3: Ablation study of link prediction performance (on the testing set) on Amazon-Movie, Amazon-Apps, Goodreads-Crime, Goodreads-Children, and StackOverflow. (-) means removing the corresponding virtual tokens.

| Model | Amazon-Movie | | Amazon-Apps | | Goodreads-Crime | | Goodreads-Children | | StackOverflow | |
|---|---|---|---|---|---|---|---|---|---|---|
| | MRR | NDCG | MRR | NDCG | MRR | NDCG | MRR | NDCG | MRR | NDCG |
| Edgeformer-N | **0.2919** | **0.4344** | 0.2239 | 0.3771 | **0.2395** | **0.3875** | **0.1446** | **0.3000** | **0.1754** | **0.3339** |
| - center node token | 0.2899 | 0.4325 | 0.2178 | 0.3717 | 0.2361 | 0.3847 | 0.1407 | 0.2964 | 0.1702 | 0.3291 |
| - neighbor node token | 0.2880 | 0.4306 | 0.2115 | 0.3656 | 0.2322 | 0.3807 | 0.1411 | 0.2963 | 0.1730 | 0.3310 |
| - neighbor text token | 0.2895 | 0.4321 | **0.2260** | **0.3789** | 0.2386 | 0.3867 | 0.1442 | 0.2998 | 0.1734 | 0.3314 |

Table 4: Node classification performance on Amazon-Movie and Amazon-App.

| Model | Amazon-Movie | | | Amazon-Apps | | |
|---|---|---|---|---|---|---|
| | Macro-F1 | Micro-F1 | PREC | Macro-F1 | Micro-F1 | PREC |
| MF | 0.7566±0.0017 | 0.8234±0.0013 | 0.8241±0.0013 | 0.4647±0.0151 | 0.8393±0.0012 | 0.8462±0.0006 |
| CensNet | 0.8528±0.0010 | 0.8839±0.0008 | 0.8845±0.0007 | 0.2782±0.0168 | 0.8279±0.0006 | 0.8331±0.0005 |
| NENN | 0.9186±0.0008 | 0.9341±0.0008 | 0.9347±0.0007 | 0.3408±0.0082 | 0.8789±0.0019 | 0.8819±0.0017 |
| BERT | 0.9209±0.0005 | 0.9361±0.0003 | 0.9367±0.0003 | 0.7608±0.0175 | 0.9283±0.0015 | 0.9337±0.0015 |
| BERT+CensNet | 0.9032±0.0006 | 0.9221±0.0004 | 0.9227±0.0004 | 0.5750±0.0277 | 0.8692±0.0034 | 0.8731±0.0028 |
| BERT+NENN | 0.9247±0.0005 | 0.9387±0.0004 | 0.9393±0.0005 | 0.7556±0.0092 | 0.9306±0.0008 | 0.9382±0.0006 |
| Edgeformer-N | **0.9276±0.0007** | **0.9411±0.0006** | **0.9417±0.0005** | **0.7758±0.0100** | **0.9339±0.0007** | **0.9431±0.0005** |

without neighbor edge information virtual token in Amazon-Apps, which indicates the importance of all three virtual token hidden states.

**Node Classification with unsupervised node embedding.** To further evaluate the quality of the unsupervised learned node embeddings, we fix the node embeddings obtained from link prediction and train a logistic regression classifier to predict nodes' categories. This is a multi-class multi-label classification task, where there are 2 classes for Amazon-Movie and 26 classes for Amazon-Apps. Table 4 summarizes the performance comparison between several edge-aware methods. We can find that: (a) Edgeformer-N can outperform all the baselines significantly and consistently, which indicates that Edgeformer-N can learn more effective node representations; (b) edge-aware models can have better performance, but it depends on how the edge text information is employed.

### 4.4 EMBEDDING VISUALIZATION

To reveal the relation between edge embeddings and node embeddings learned by our model, we apply t-SNE (Van der Maaten & Hinton, 2008) to visualize them in Figure 2. Node embeddings (*i.e.*, $\{h_v | v \in \mathcal{V}\}$) are denoted as stars, while edge embeddings (*i.e.*, $\{h_e | e \in \mathcal{E}\}$) are denoted as points with the same color as the node they link to. From the figure, we observe that: (1) node embeddings tend to be closer to each other in the embedding space compared with edge embeddings; (2) the embeddings of edges linked to the same node are in the vicinity of each other.

### 4.5 EFFICIENCY ANALYSIS

We now compare the efficiency of BERT+GIN (a node-centric GNN), BERT+NENN (an edge-aware GNN), GraphFormers (a PLM-GNN nested architecture), and our Edgeformer-N. All models are run on one NVIDIA A6000. The mini-batch size is 25; each sample contains one center node and $|\mathcal{N}_e(v)|$ neighbor edges; the maximum text length is 64 tokens. The running time (per mini-batch) of compared models is reported in Table 5, where we have the following findings: (a) the time cost of training Edgeformer-N is quite close to that of BERT+GIN, BERT+NENN, and GraphFormers; (b) PLM-GNN nested architectures (i.e., GraphFormers and Edgeformer-N) require slightly longer time during training than PLM-GNN cascaded architectures (i.e., BERT+GIN and BERT+NENN); (c) the time cost of Edgeformer-N increases linearly with the neighbor size $|\mathcal{N}_e(v)|$, which is consistent with our analysis in Section 3.3 that the time complexity of Edgeformer-N is $\mathcal{O}(NP^2 + N^2) \sim \mathcal{O}(NP^2)$ when $N \ll P$.

## 5 RELATED WORK

### 5.1 PRETRAINED LANGUAGE MODELS

PLMs are proposed to learn universal language representations from large-scale text corpora. Early studies such as word2vec (Mikolov et al., 2013), fastText (Bojanowski et al., 2017), and GloVe (Pennington et al., 2014) aim to learn a set of context-independent word embeddings to capture word semantics. However, many NLP tasks are beyond word-level, so it is beneficial to derive word representations based on specific contexts. Contextualized language models are extensively studied

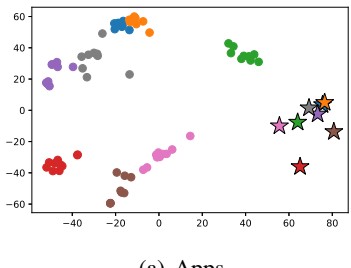 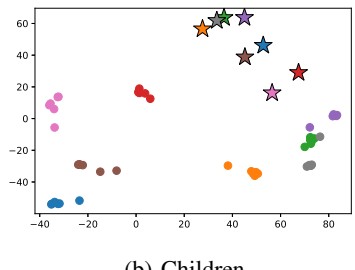

(a) Apps                                  (b) Children

Figure 2: Embedding visualization. Node embeddings are denoted as stars, and the embeddings of edges are denoted as points with the same color if they are linked to the same node.

Table 5: Time cost (*ms*) per mini-batch for BERT+GIN, BERT+NENN, GraphFormers, and Edgeformer-N, with neighbor size $|\mathcal{N}_e(v)|$ increasing from 2 to 5 on Amazon-Apps, Goodreads-Children, and StackOverflow. Edgeformer-N achieves similar efficiency with the baselines.

| Model | Amazon-Apps | | | | Goodreads-Children | | | | StackOverflow | | | |
|---|---|---|---|---|---|---|---|---|---|---|---|---|
| | 2 | 3 | 4 | 5 | 2 | 3 | 4 | 5 | 2 | 3 | 4 | 5 |
| BERT+GIN | 15.53 | 21.44 | 25.69 | 31.19 | 15.44 | 21.38 | 25.88 | 31.02 | 15.29 | 21.41 | 25.58 | 31.05 |
| BERT+NENN | 15.70 | 21.71 | 26.03 | 31.50 | 15.78 | 21.86 | 26.15 | 31.46 | 15.74 | 21.97 | 26.09 | 31.46 |
| GraphFormers | 17.21 | 23.56 | 28.29 | 34.43 | 17.08 | 23.76 | 28.43 | 34.38 | 17.13 | 23.65 | 28.60 | 34.45 |
| Edgeformer-N | 18.68 | 25.39 | 30.45 | 36.57 | 18.74 | 25.17 | 30.42 | 36.42 | 18.57 | 25.32 | 30.46 | 36.32 |

recently to achieve this goal. For example, GPT (Peters et al., 2018; Radford et al., 2019) adopts auto-regressive language modeling to predict a token given all previous tokens; BERT (Devlin et al., 2019) and RoBERTa (Liu et al., 2019) are trained via masked language modeling to recover randomly masked tokens; XLNet (Yang et al., 2019) proposes permutation language modeling; ELECTRA (Clark et al., 2020) uses an auxiliary Transformer to replace some tokens and pretrains the main Transformer to detect the replaced tokens. For more related studies, one can refer to a recent survey (Qiu et al., 2020). To jointly leverage text and graph information, previous studies (Zhang et al., 2019; Fang et al., 2020; Li et al., 2021; Zhu et al., 2021) propose a PLM-GNN cascaded architecture, where the text information of each node is first encoded via PLMs, then the node representations are aggregated via GNNs. (Bi et al., 2022) proposes a triple2seq operation to linearize subgraphs and a "mask prediction" paradigm to conduct inference. Recently, GraphFormers (Yang et al., 2021) introduces a GNN-nested Transformer to stack GNN layers and Transformer layers alternately. However, these works mainly consider *textual-node* networks, thus their focus is orthogonal to ours on *textual-edge* networks.

### 5.2 EDGE-AWARE GRAPH NEURAL NETWORKS

A vast majority of GNN models (Kipf & Welling, 2017; Hamilton et al., 2017; Velickovic et al., 2018; Xu et al., 2019) leverage node attributes only and lack specific designs to utilize edge features. Heterogeneous GNNs (Schlichtkrull et al., 2018; Yang et al., 2020) assume each edge has a pre-defined type and take such types into consideration during aggregation. However, they still cannot deal with more complicated features (*e.g.*, text) associated with the edges. EGNN (Gong & Cheng, 2019) introduces an attention mechanism to inject edge features into node representations; CensNet (Jiang et al., 2019) alternately updates node embeddings and edge embeddings in convolution layers; NENN (Yang & Li, 2020) aggregates the representation of each node/edge from both its node and edge neighbors via a GAT-like attention mechanism. EHGNN (Jo et al., 2021) proposes the dual hypergraph transformation and conducts graph convolutions for edges. Nevertheless, these models do not collaborate PLMs and GNNs to specifically deal with text features on edges, thus they underperform our Edgeformers model, even stacked with a BERT encoder.

### 6 CONCLUSIONS

We tackle the problem of representation learning on textual-edge networks. To this end, we propose a novel graph-empowered Transformer framework, which integrates local network structure information into each Transformer layer text encoding for edge representation learning and aggregates edge representation fused by network and text signals for node representation. Comprehensive experiments on five real-world datasets from different domains demonstrate the effectiveness of Edgeformers on both edge-level and node-level tasks. Interesting future directions include (1) exploring other variants of introducing network signals into Transformer text encoding and (2) applying the framework to more network-related tasks such as recommendation and text-rich social network analysis.

ACKNOWLEDGMENTS

We thank anonymous reviewers for their valuable and insightful feedback. Research was supported in part by US DARPA KAIROS Program No. FA8750-19-2-1004 and INCAS Program No. HR001121C0165, National Science Foundation IIS-19-56151, IIS-17-41317, and IIS 17-04532, and the Molecule Maker Lab Institute: An AI Research Institutes program supported by NSF under Award No. 2019897, and the Institute for Geospatial Understanding through an Integrative Discovery Environment (I-GUIDE) by NSF under Award No. 2118329. Any opinions, findings, and conclusions or recommendations expressed herein are those of the authors and do not necessarily represent the views, either expressed or implied, of DARPA or the U.S. Government.

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

# A  APPENDIX

## A.1  DATASETS

The statistics of the five datasets can be found in Table 6.

Table 6: Dataset Statistics

| Dataset | # Node | # Edge |
|---|---|---|
| Amazon-Movie | 173,986 | 1,697,533 |
| Amazon-Apps | 100,468 | 752,937 |
| Goodreads-Crime | 385,203 | 1,849,236 |
| Goodreads-Children | 192,036 | 734,640 |
| StackOverflow | 129,322 | 281,657 |

## A.2  SUMMARY OF EDGEFORMERS' ENCODING PROCEDURE

---

**Algorithm 1:** Edge Representation Learning Procedure of Edgeformer-E

---

**Input** : The edge $e_{ij}$, its associated text $d_{ij}$, and its involved nodes $v_i$ and $v_j$.
The initial token embeddings $\boldsymbol{H}_{e_{ij}}^{(0)}$ of the document $d_{ij}$.

**Output :** The embedding $\boldsymbol{h}_{e_{ij}}$ of the edge $e_{ij}$.

**begin**

   // obtain layer-1 representation of text tokens

   $\boldsymbol{H}_{e_{ij}}^{(0)'} \leftarrow \text{Normalize}(\boldsymbol{H}_{e_{ij}}^{(0)} + \text{MHA}^{(0)}(\boldsymbol{H}_{e_{ij}}^{(0)}))$ ;

   $\boldsymbol{H}_{e_{ij}}^{(1)} \leftarrow \text{Normalize}(\boldsymbol{H}_{e_{ij}}^{(0)'} + \text{FFN}^{(0)}(\boldsymbol{H}_{e_{ij}}^{(0)'}))$ ;

   // obtain the base embedding of $v_i$ and $v_j$

   $\boldsymbol{z}_{v_i}^{(0)} \leftarrow \text{Embedding}(v_i)$ ;

   $\boldsymbol{z}_{v_j}^{(0)} \leftarrow \text{Embedding}(v_j)$ ;

   **for** $l = 1, ..., L$ **do**

      // obtain layer-$l$ representation of virtual node tokens

      $\boldsymbol{z}_{v_i}^{(l)} \leftarrow \boldsymbol{W}_n^{(l)} \boldsymbol{z}_{v_i}^{(0)}$ ;

      $\boldsymbol{z}_{v_j}^{(l)} \leftarrow \boldsymbol{W}_n^{(l)} \boldsymbol{z}_{v_j}^{(0)}$ ;

      // obtain layer-$(l+1)$ representation of text tokens

      $\widetilde{\boldsymbol{H}}_{e_{ij}}^{(l)} = \boldsymbol{z}_{v_i}^{(l)} \parallel \boldsymbol{z}_{v_j}^{(l)} \parallel \boldsymbol{H}_{e_{ij}}^{(l)}$ ;

      $\boldsymbol{H}_{e_{ij}}^{(l)'} \leftarrow \text{Normalize}(\boldsymbol{H}_{e_{ij}}^{(l)} + \text{MHA}_{asy}^{(l)}(\boldsymbol{H}_{e_{ij}}^{(l)}, \widetilde{\boldsymbol{H}}_{e_{ij}}^{(l)}))$ ;

      $\boldsymbol{H}_{e_{ij}}^{(l+1)} \leftarrow \text{Normalize}(\boldsymbol{H}_{e_{ij}}^{(l)'} + \text{FFN}^{(l)}(\boldsymbol{H}_{e_{ij}}^{(l)'}))$ ;

   **end**

   **return** $\boldsymbol{h}_{e_{ij}} \leftarrow \boldsymbol{H}_{e_{ij}}^{(L+1)}[\text{CLS}]$ ;

**end**

---

## A.3  EDGE CLASSIFICATION

We also compare our method with the state-of-the-art edge representation learning method EHGNN (Jo et al., 2021) and two node-centric PLM-GNN methods BERT+MaxSAGE (Hamilton et al., 2017) and GraphFormers (Yang et al., 2021). The experimental results can be found in Table 7. From the results, we can find that Edgeformer-E consistently outperforms all the baseline methods, including EHGNN, BERT+EHGNN, BERT+MaxSAGE and GraphFormers.

EHGNN cannot obtain promising results because of two reasons: 1) edge-edge propagation: EHGNN proposes to transform edge to node and node to edge in the original network, followed by graph convolutions on the new hypernetwork. This results in edge-edge information propagation when conducting edge representation learning. However, edge-edge information propagation has the underlying edge-edge homophily assumption which is not always true in textual-edge networks. For example, when predicting the rate of a review text $e_{ij}$ given by $u$ to $i$, it is not straightforward to make the judgment based on reviews for $i$ written by other users (neighbor edges); 2) The integration

---

**Algorithm 2:** Node Representation Learning Procedure of Edgeformer-N.

---

**Input** : The center node $v_i$, its edge neighbors $\mathcal{N}_e(v_i)$, and its node neighbors $\mathcal{N}_n(v_i)$.

         The initial token embedding $\boldsymbol{H}_{e_{ij}}^{(0)}$ of each document $d_{ij}$ associated with $e_{ij} \in \mathcal{N}_e(v_i)$.

**Output :** The embedding $\boldsymbol{h}_{v_i}$ of the center node $v_i$.

**begin**

   // obtain layer-1 representation of text tokens

   **for** $e_{ij} \in N_e(v_i)$ **do**

      $\boldsymbol{H}_{e_{ij}}^{(0)'} \leftarrow \text{Normalize}(\boldsymbol{H}_{e_{ij}}^{(0)} + \text{MHA}^{(0)}(\boldsymbol{H}_{e_{ij}}^{(0)}))$ ;

      $\boldsymbol{H}_{e_{ij}}^{(1)} \leftarrow \text{Normalize}(\boldsymbol{H}_{e_{ij}}^{(0)'} + \text{FFN}^{(0)}(\boldsymbol{H}_{e_{ij}}^{(0)'}))$ ;

   **end**

   // obtain the base embedding of each node

   **for** $u \in N_n(v_i) \cup \{v_i\}$ **do**

      $\boldsymbol{z}_u^{(0)} \leftarrow \text{Embedding}(u)$ ;

   **end**

   **for** $l = 1, ..., L$ **do**

      // obtain layer-$l$ representation of virtual node tokens

      **for** $u \in N_n(v_i) \cup \{v_i\}$ **do**

         $\boldsymbol{z}_u^{(l)} \leftarrow \boldsymbol{W}_n^{(l)} \boldsymbol{z}_u^{(0)}$ ;

      **end**

      // obtain layer-$l$ representation of virtual neighbor aggregation tokens

      **for** $e_{ij} \in N_e(v_i)$ **do**

         $\boldsymbol{h}_{e_{ij}}^{(l)} \leftarrow \boldsymbol{H}_{e_{ij}}^{(l)}[\text{CLS}]$ ;

      **end**

      $\left[\bar{\boldsymbol{h}}_{e_{ij}|v_i}^{(l)}, ..., \bar{\boldsymbol{h}}_{e_{is}|v_i}^{(l)}\right] \leftarrow \text{MHA}_g^{(l)}\left(\left[\boldsymbol{h}_{e_{ij}}^{(l)}, ..., \boldsymbol{h}_{e_{is}}^{(l)}\right]\right)$ ;

      // obtain layer-$(l+1)$ representation of text tokens

      **for** $e_{ij} \in N_e(v_i)$ **do**

         $\widetilde{\boldsymbol{H}}_{e_{ij}}^{(l)} \leftarrow \boldsymbol{z}_i^{(l)} \parallel \boldsymbol{z}_j^{(l)} \parallel \bar{\boldsymbol{h}}_{e_{ij}|v_i}^{(l)} \parallel \boldsymbol{H}_{e_{ij}}^{(l)}$ ;

         $\boldsymbol{H}_{e_{ij}}^{(l)'} \leftarrow \text{Normalize}(\boldsymbol{H}_{e_{ij}}^{(l)} + \text{MHA}_{asy}^{(l)}(\boldsymbol{H}_{e_{ij}}^{(l)}, \widetilde{\boldsymbol{H}}_{e_{ij}}^{(l)}))$ ;

         $\boldsymbol{H}_{e_{ij}}^{(l+1)} \leftarrow \text{Normalize}(\boldsymbol{H}_{e_{ij}}^{(l)'} + \text{FFN}^{(l)}(\boldsymbol{H}_{e_{ij}}^{(l)'}))$ ;

      **end**

   **end**

   // obtain the edge representation

   **for** $e_{ij} \in N_e(v_i)$ **do**

      $\boldsymbol{h}_{e_{ij}|v_i} \leftarrow \boldsymbol{H}_{e_{ij}}^{(L+1)}[\text{v}_\text{j}]$ ;

   **end**

   // obtain the node representation

   $\boldsymbol{h}_{v_i} = \text{AGG}(\{\boldsymbol{h}_{e_{ij}|v_i}|e_{ij} \in N_e(v_i)\})$ ;

   **return** $\boldsymbol{h}_{v_i}$

**end**

---

of text and network signals are loose for BERT+EHGNN, since such architectures process text and graph signals one after the other, and fail to simultaneously model the deep interactions between both types of information. However, our Edgeformer-E is designed following the more reasonable node-edge homophily hypothesis and deeply integrating text & network signals by introducing virtual node tokens in Transformer encoding.

Note that both PLM+GNN and Edgeformer-E require textual information on ALL nodes in the network. However, this assumption does not hold in many textual-edge networks. Therefore, we propose a way around to concatenate the text on the edges linked to the given node together to make up node text. However, such a strategy does not lead to competitive performance of PLM+MaxSAGE and GraphFormers according to our experimental results. Therefore, to make our model generalizable to the case of missing node text, the proposed Edgeformers can be a better solution.

Table 7: Edge classification performance on Amazon-Movie, Amazon-App, Goodreads-Crime, and Goodreads-Children.

| Model | Amazon-Movie | | Amazon-Apps | | Goodreads-Crime | | Goodreads-Children | |
|---|---|---|---|---|---|---|---|---|
| | Macro-F1 | Micro-F1 | Macro-F1 | Micro-F1 | Macro-F1 | Micro-F1 | Macro-F1 | Micro-F1 |
| TF-IDF | 50.01 | 64.22 | 48.30 | 62.88 | 43.07 | 51.72 | 39.42 | 49.90 |
| TF-IDF+nodes | 53.59 | 66.34 | 50.56 | 65.08 | 49.35 | 57.50 | 47.32 | 56.78 |
| EHGNN | 49.90 | 64.04 | 48.20 | 63.63 | 44.49 | 52.30 | 40.01 | 50.23 |
| BERT | 61.38 | 71.36 | 59.11 | 69.27 | 56.41 | 61.29 | 51.57 | 57.72 |
| BERT+nodes | *63.00* | *72.45* | *59.72* | *70.82* | *58.64* | *65.02* | *54.42* | *60.46* |
| BERT+EHGNN | 61.45 | 70.73 | 58.86 | 70.79 | 56.92 | 61.66 | 52.46 | 57.97 |
| BERT+MaxSAGE | 61.57 | 70.79 | 58.95 | 70.45 | 57.20 | 61.98 | 52.75 | 58.53 |
| GraphFormers | 61.73 | 71.52 | 59.67 | 70.19 | 57.49 | 62.37 | 52.93 | 58.34 |
| Edgeformer-E | **64.18** | **73.59** | **60.67** | **71.28** | **61.03** | **65.86** | **57.45** | **61.71** |

## A.4 LINK PREDICTION

We further report the link prediction performance of compared models on the validation set in Table 8.

Table 8: Link prediction performance (on the validation set) on Amazon-Movie, Amazon-Apps, Goodreads-Crime, Goodreads-Children, and StackOverflow. $\Delta$ denotes the relative improvement of our model comparing with the best baseline.

| Model | Amazon-Movie | | Amazon-Apps | | Goodreads-Crime | | Goodreads-Children | | StackOverflow | |
|---|---|---|---|---|---|---|---|---|---|---|
| | MRR | NDCG | MRR | NDCG | MRR | NDCG | MRR | NDCG | MRR | NDCG |
| MF | 0.2178 | 0.3666 | 0.1523 | 0.3086 | 0.2492 | 0.3966 | 0.1470 | 0.3042 | 0.1104 | 0.2702 |
| MeanSAGE | 0.2280 | 0.3775 | 0.1804 | 0.3375 | 0.2286 | 0.3792 | 0.1348 | 0.2927 | 0.1258 | 0.2846 |
| MaxSAGE | 0.2321 | 0.3812 | 0.1708 | 0.3288 | 0.2299 | 0.3812 | 0.1339 | 0.2919 | 0.1257 | 0.2848 |
| GIN | 0.2287 | 0.3769 | 0.1846 | 0.3402 | 0.2306 | 0.3802 | 0.1420 | 0.2989 | 0.1275 | 0.2860 |
| CensNet | 0.2186 | 0.3682 | 0.1953 | 0.3504 | 0.2399 | 0.3875 | 0.1501 | 0.3059 | 0.1338 | 0.2900 |
| NENN | 0.2776 | 0.4204 | 0.2068 | 0.3610 | 0.2777 | 0.4224 | 0.1658 | 0.3207 | 0.1361 | 0.2948 |
| BERT | 0.2582 | 0.4017 | 0.1863 | 0.3407 | 0.2540 | 0.4001 | 0.1608 | 0.3156 | 0.1798 | 0.3371 |
| BERT+MaxSAGE | 0.3028 | 0.4424 | 0.2128 | 0.3661 | 0.2859 | 0.4299 | 0.1687 | 0.3236 | 0.1828 | 0.3399 |
| BERT+MeanSAGE | 0.2705 | 0.4145 | 0.2024 | 0.3572 | 0.2527 | 0.4004 | 0.1572 | 0.3129 | 0.1849 | *0.3418* |
| BERT+GIN | 0.2790 | 0.4212 | 0.2040 | 0.3583 | 0.2613 | 0.4073 | 0.1611 | 0.3158 | *0.1858* | 0.3413 |
| GraphFormers | 0.2998 | 0.4393 | 0.2111 | 0.3642 | 0.2852 | 0.4294 | 0.1671 | 0.3220 | 0.1833 | 0.3405 |
| BERT+CensNet | 0.2025 | 0.3552 | 0.1577 | 0.3163 | 0.1822 | 0.3361 | 0.1007 | 0.2603 | 0.1232 | 0.2845 |
| BERT+NENN | *0.3087* | *0.4470* | *0.2193* | *0.3719* | *0.2956* | *0.4382* | *0.1737* | *0.3280* | 0.1759 | 0.3341 |
| Edgeformer-N | **0.3206** | **0.4574** | **0.2320** | **0.3838** | **0.3106** | **0.4514** | **0.1849** | **0.3385** | **0.1944** | **0.3508** |
| + $\Delta$ % | **3.9%** | **2.3%** | **5.8%** | **3.2%** | **5.1%** | **3.0%** | **6.4%** | **3.2%** | **4.6%** | **2.6%** |

## A.5 NODE CLASSIFICATION

The 26 classes of Amazon-Apps nodes are: "Books & Comics", "Communication", "Cooking", "Education", "Entertainment", "Finance", "Games", "Health & Fitness", "Kids", "Lifestyle", "Music", "Navigation", "News & Magazines", "Novelty", "Photography", "Podcasts", "Productivity", "Reference", "Ringtones", "Shopping", "Social Networking", "Sports", "Themes", "Travel", "Utilities", and "Weather".

The 2 classes of Amazon-Movie nodes are: "Movies" and "TV".

## A.6   HYPER-PARAMETER STUDY

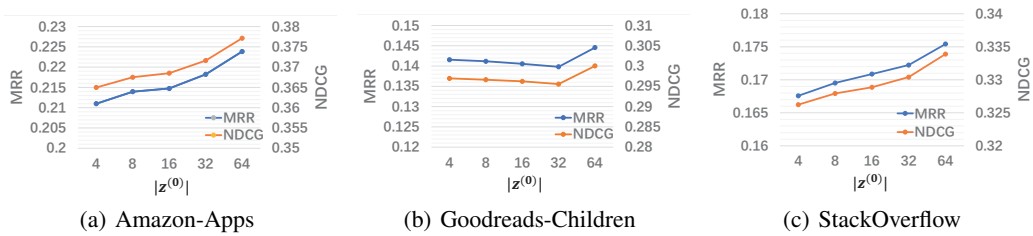

| (a) Amazon-Apps | (b) Goodreads-Children | (c) StackOverflow |

Figure 3: Effect of the dimension of initial node embeddings.

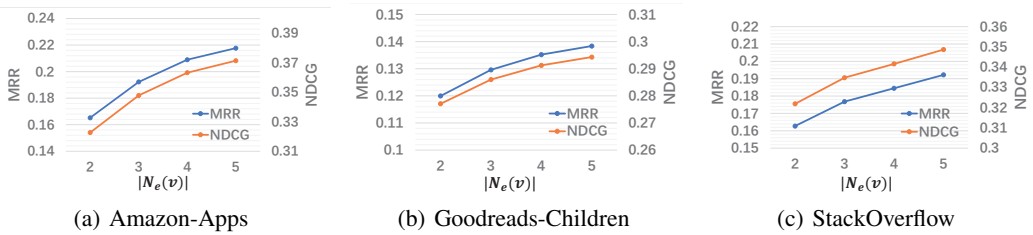

| (a) Amazon-Apps | (b) Goodreads-Children | (c) StackOverflow |

Figure 4: Effect of the sampled neighbor size (*i.e.*, $|N_e(v)|$).

**Node Dimension.**   We conduct experiments on Amazon-Apps, Goodreads-Children, and Stack-Overflow to understand the effect of the initial node embedding dimension in Eq.(9). We test the performance of Edgeformer-N on the link prediction task with the initial node embedding dimension varying in 4, 8, 16, 32, and 64. The results are shown in Figure 3, where we can find that the performance of Edgeformer-N generally increases as the initial node embedding dimension increases. This finding is straightforward since the more parameters an initial node embedding has before overfitting, the more information it can represent.

**Sampled Neighbor Size.**   We further analyze the impact of sampled neighbor size for node representation learning on Amazon-Apps, Goodreads-Children, and StackOverflow, with a fraction of edges randomly sampled for the center node. The result can be found in Figure 4. We can find that the performance increases progressively as sampled neighbor size $|N_e(v)|$ increases. It is intuitive since the more neighbors we have, the more information can contribute to center node learning. Meantime, the increase rate decreases as $|N_e(v)|$ increases linearly because the information between neighbors can have information overlap.

## A.7   SELF-ATTENTION MAP STUDY

In order to study how the virtual node token will benifit the encoding of Edgeformer-E, we plot the self-attention probability map for a random sample in Figure 5. We random pick up a token from this sample and plot the self-attention probability of how different tokens (x-axis), including virtual node tokens and the first twenty original text tokens, will contribute to the encoding of this random token in different layers (y-axis). From the figure, we can find that: In higher layers (e.g., Layers 10-11), the attention weights of virtual node tokens are significantly larger than those of original node tokens. Since virtual node token hidden states are of $R^{d \times 2}$ and the original text token hidden states are of $R^{d \times l}$ ($l$ is text sequence length), the ratio of network tokens to text tokens is $2 : l$ in $\widetilde{H}_{e_{ij}}^{(l)}$ (Eq.4), where $l$ is the text sequence length. However, the self-attention mechanism can automatically learn to balance the two types of information by assigning higher weights to the corresponding virtual node tokens, so a larger number of tokens representing textual information will not cause network information to be overwhelmed.

## A.8   REPRODUCIBILITY SETTINGS

For a fair comparison, the training objectives of Edgeformer-N and all PLM-involved baselines are the same. The hyper-parameter configuration for obtaining the results in Tables 1 and 2 can be found

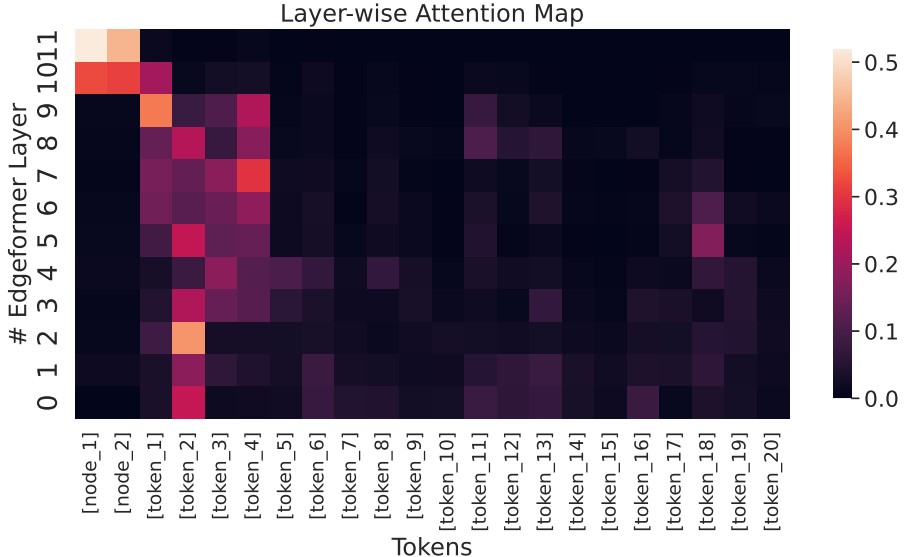

Figure 5: Self-attention probability map of Edgeformer-E for a random sample. The x-axis corresponds to different key/value tokens and the y-axis corresponds to different Edgeformer-E layers. In higher layers (e.g., layers 10-11), the attention weights of virtual node tokens are significantly larger than those of original node tokens. The ratio of network tokens to text tokens is $2 : l$ in $\widetilde{\boldsymbol{H}}_{e_{ij}}^{(l)}$ (Eq.4), where $l$ is the text sequence length. However, the self-attention mechanism can automatically learn to balance the two types of information by assigning higher weights to the corresponding virtual node tokens, so a larger number of tokens representing textual information will not cause network information to be overwhelmed.

in Table 9, where "sampled neighbor size" stands for the number of neighbors sampled for each type of the center node during node representation learning. This hyper-parameter is determined according to the average node degree of the corresponding node type. The edge classification and link prediction experiments are conducted on one NVIDIA V100 and one NVIDIA A6000, respectively.

In Section 4.3, we adopt logistic regression as our classifier. We employ the Adam optimizer (Kingma & Ba, 2015) with the early-stopping patience as 10 to train our classifier. The learning rate is set as 0.001.

## A.9 LIMITATIONS

In this work, we mainly focus on modeling homogeneous textual-edge networks and solving fundamental tasks in graph learning such as node/edge classification and link prediction. Interesting future studies include designing models to characterize network heterogeneity and applying our proposed model to real-world applications such as recommendation.

Table 9: Hyper-parameter configuration.

| Parameter | Amazon-Movie | Amazon-Apps | Goodreads-Crime | Goodreads-Children | StackOverflow |
|---|---|---|---|---|---|
| optimizer | | | Adam | | |
| learning rate | | | 1e-5 | | |
| weight decay | | | 1e-3 | | |
| Adam $\epsilon$ | | | 1e-8 | | |
| early-stopping patience | | | 3 | | |
| training batch size | | | 30 | | |
| testing batch size | | | 100 | | |
| node embedding dim | | | 64 | | |
| chunk $k$ | | | 12 | | |
| max sequence length | | | 64 | | |
| backbone PLM | | | BERT-base-uncased | | |
| sampled neighbor size | user:8 item:10 | user:3 item:5 | reader:8 book:10 | reader:6 book:4 | expert:2 question:5 |

## A.10 ETHICAL CONSIDERATIONS

While it has been demonstrated that PLMs are powerful in language understanding (Devlin et al., 2019; Liu et al., 2019; Clark et al., 2020), there are studies pointing out their drawbacks such as containing social bias (Liang et al., 2021) and misinformation (Abid et al., 2021). In our work, we focus on enriching PLMs' text encoding process with the associated network structure information, which could be a way to mitigate the bias and wipe out the contained misinformation.

