# OpenReview forum: "Edgeformers: Graph-Empowered Transformers for Representation Learning on Textual-Edge Networks"
_ICLR.cc/2023/Conference — ICLR 2023 poster_

### Official Review · Reviewer_CnjZ · 2022-10-19

**Confidence:** 3
**Correctness:** 4
**Technical Novelty And Significance:** 2
**Empirical Novelty And Significance:** 3
**Recommendation:** 8

**Clarity, Quality, Novelty And Reproducibility:**

The paper is mostly clear, even though I had to take a look at the appendices to have a clearer understanding of the final procedure (Alg.1&2). The authors give access to their code and the paper contains many details about the experimental setting.

I think the novelty is quite limited because the authors combine already well-known techniques. There is no new idea nor insight that can foster further research.

We can find many details of the implementation so that the experiments can be reproduced. As I said earlier, it's surprising that the usual information about how data have been split into training/test sets is not given.

**Details Of Ethics Concerns:**

-

**Strength And Weaknesses:**

Strength:

- dealing with the textual information in a proper, modern way
- experiments on various and quite big datasets with positive, consistant results

Weaknesses:

- The architecture is quite complex with multiple mechanisms that are combined and it's not always easy to figure out what's really going on in there. For instance, Edgeformer-E integrates node vectors noted z_vi (for vertices vi) in order to contextualized the edge vector h_eij. However, Edgeformer-N uses another node vector noted h_vi which is the aggregation of the surrounding h_eij. In this end, I wonder what's the relation between both types of vectors. I guess that z_vi is a *first version* that has been a priori initialized with another node embedding technique, and h_eij a *second (and better) version* that is maybe more contextualized.
- Some recent works have pointed out the bias of GNNs that tend to "oversmooth" the information because of the propagation mechanism (see for instance the discussion here: https://towardsdatascience.com/over-smoothing-issue-in-graph-neural-network-bddc8fbc2472). It seems to me that EdgeFormer has the same king of limitation. If I'm correct, I think the authors should discuss this point and test on other datasets to clearly show when their model is applicable.
- Even though the authors make some effort for reproducibility, I don't see any information related to the usual train/test procedure we expect in a machine learning paper.
- It seems to me that the framework is transductive. What would be its ability for more inductive tasks? For instance, what about a new node (e.g., a new user) that posts a review on a product which is in the network? Is EdgeFormer able to deal with such a case?

Typo:
- I think "z_i" and "z_j" should be replaced by "z_vi" and "z_vj" in Eq.12

**Summary Of The Paper:**

EdgeFormers is a new deep learning architecture to deal with networks with textual edges (e.g., email network or product reviews). Actually, it integrates two sequential modules: the first module (Edgeformer-E) aims at learning edge text-aware representations, the second module (Edgeformer-N) aims at learning node representation based on the edge representations of its neighbors. This work doesn't deal with attributed nodes, even though it should be easily adapted to do so. The main point is to go beyond considering the information associated to edges as a set of features, but treating it as a full text by the mean of Transformer-like mechanisms (multihead attention). EdgeFormers is trained on both edge classification (for Edgeformer-E module) and link prediction (for Edgeformer-N module). It's favorably compared to several recent architectures on 5 datasets.

**Summary Of The Review:**

The proposed architecture is not fundamentally novel but it seems more fitted to deal with networks with textual edges. The model is quite well designed, even though some parts should be made clearer (relation between h_vi and z_vi, see above). EdgeFormer leads to slightly better results on 5 datasets, which finally explains my recommandation.

---

> ### Author Response · Authors · 2022-11-15
> **Response to Reviewer CnjZ**
>
> Thank you so much for your thoughtful review!
>
> Regarding your questions:
>
> **1. $z_{v_i}$ & $h_{e_{ij}|v_{i}}$.** Your understanding is correct. $z_{v_i}$ can be seen as a prior representation of node $v_i$, while $h_{e_{ij}|v_{i}}$ is a further contextualized representation given $v_i$’s local network information. For example, user $v_i$ (a CS professor) and user $v_j$ (a CS student) are emailing text $e_{ij}$ about “Transformers”. We also know $v_i$’s other emails $e_{ik}$ about “Machine Learning”. From “a CS professor”, “a CS student” and “Machine Learning”, we can understand that “Transformers” in $e_{ij}$ given $v_i$ refers to a deep learning architecture rather than a character in the movie. Here, “a CS professor” -> $z_{v_i}$, “a CS student” -> $z_{v_j}$, “Machine Learning” -> $h_{e_{ij}|v_{i}}$.
>
> **2. Oversmoothing.** Oversmoothing is a problem for GNN, that when the GNN layers go deeper, the final representations of nodes in the network will be indistinguishable because of information propagation. However, we argue that Edgeformers are not exposed to oversmoothing:
> - Edgeformers are different from GNN: even though there are some connections between Edgeformers and GNN, they have completely different mechanisms. GNN relies mainly on network information propagation, which is the cause of oversmoothing. Nevertheless, Edgeformers mainly focus on text encoding with network information as extra signals. When Edgeformers go deeper, it means the integration of text and network signal is tighter, which is different from GNN (deeper GNN means that each node can access its far away neighbors and obtain information propagated from them which will finally cause over-smoothing on the whole network).
> - Edgeformers mainly rely on 1-hop neighbors: Even though Edgeformers layer goes deeper, it still only leverages 1-hop neighbor information, which is congenitally resistant to oversmoothing.
>
> **3. Dataset and data processing.** Thanks for pointing it out. We have uploaded all the data processing code to https://anonymous.4open.science/r/Edgeformer-release-F422. The train/val/test split is 7:1:2 for all experiments. The processed data can be generated with the data processing code and is uploaded to https://gofile.io/d/oqUDoW.
>
> **4. Transductive or inductive.** This is a very good point. Currently, due to the definition of the problem, the whole framework is transductive. Since nodes are not associated with attribute information, we need to learn a base embedding $z^0_{v_i}$ for each node $v_i$ in the network. After training, we can obtain $z^0_{v_i}$, but they are specific to nodes seen in the training network. (When a new node $v_j$ comes, we need to train the model again in order to obtain  $z^0_{v_j}$.) In this case, the whole model is transductive. However, our model requires minimal changes to be applicable to inductive settings. When nodes are naturally associated with attribute information, we can use each node’s attribute vector as $z^0_{v_i}$ instead of learning a base embedding using the network. In this case, there is no model component specific to the training network. As a result, when a new node comes, by feeding its attribute vector and neighbor information into our model, one can directly obtain the prediction results.

---

### Official Review · Reviewer_Sciz · 2022-10-25

**Confidence:** 4
**Clarity, Quality, Novelty And Reproducibility:** Overall the quality, clarity and nove…
**Correctness:** 3
**Technical Novelty And Significance:** 3
**Empirical Novelty And Significance:** 3
**Recommendation:** 6

**Strength And Weaknesses:**

Overall this paper is clearly writing, well motivated. The method proposed is substantial and the results are convincing.

That said, there is still some issues that are better addressed:

1. A lot of similar works have been researched on the graph with text attributes, like PLM-GNN and GraphFormers mentioned in the introduction. Although for them the text attributes are on the node, a straightforward adaptation of these methods to the current setting would be creating an extra node with text attribute for each edge. It's unclear how do they perform and compare with the proposed method

2. In ablation study, are the differences statistically significant?

3. in equation (4), since $z$s are of $R^d$ while $H$ is of $R^{d \times n}$, are such concatenations "balancing"? (e.g. would $H$ dominates the final representation?)


**Summary Of The Paper:**

In this paper the authors address the problem of graph representation learning where edges are associated with text. In doing so the authors propose to combine pretrained networks with GNN with local network aggregation to inject text representation of nodes to its neighbors. The authors conduct experiments on various datasets and tasks, along with visualization, to show the performance of the proposed method.


**Summary Of The Review:**


Since the strengths are dominant, I would recommend 6: marginally above the acceptance threshold

---

> ### Author Response · Authors · 2022-11-15
> **Response to Reviewer Sciz**
>
> Thank you so much for your thoughtful review!
>
> Regarding your questions:
>
> **1. Converting each textual edge to a textual node.** We agree that this is an applicable and straightforward idea.
> - For the link prediction task, we have implemented this idea. In fact, the performance of PLM-GNN and GraphFormers reported in Table 2 is based on the network where each textual edge is converted to a textual node;
> - For the edge classification task, we add some experiments for the best-performing PLM-GNN baseline (i.e., PLM+MaxSAGE) and GraphFormers by using this idea.
> The Macro-F1 results and Micro-F1 results are shown in the following tables respectively (also available in Appendix A.3),
>
> | Model        | Movie | Apps  | Crime | Children |
> |--------------|-------|-------|-------|----------|
> | BERT+MaxSAGE        | 61.57 | 58.95 | 57.20 | 52.75    |
> | GraphFormers | 61.73 | 59.67 | 57.49 | 52.93    |
> | Edgeformer-E | 64.18 | 60.67 | 61.03 | 57.45    |
> | Model        | Movie | Apps  | Crime | Children |
> | BERT+MaxSAGE        | 70.79 | 70.45 | 61.98 | 58.53    |
> | GraphFormers | 71.52 | 70.19 | 62.37 | 58.34    |
> | Edgeformer-E | 73.59 | 71.28 | 65.86 | 61.71   |
>
> From the results, we can find that Edgeformer-E consistently outperforms all the baseline methods, including the newly added PLM+MaxSAGE and GraphFormers. Note that both PLM+GNN and Edgeformer-E require textual information on _ALL_ nodes in the network. However, this assumption does not hold in many textual-edge networks (e.g., many users are not associated with text information and some products have no descriptions). Therefore, we adopt a way around to concatenate the text on the edges linked to the given node together to make up node text. However, such a strategy does not lead to competitive performance of PLM+MaxSAGE and GraphFormers according to our experimental results. Therefore, to make our model generalizable to the case of missing node text, the proposed Edgeformers can be a better solution.
>
> **2. Significance in the ablation study.** The absolute standard deviation of results in each cell in the ablation study is within ±0.0010 for three repeat experiments. For example, the MRR range of “- center node token” on Goodreads-Children is 0.1407±0.0009.
>
> **3.Whether the concatenations are balancing or not.** This is a very interesting point. To answer this question, we add a section in Appendix A.7 to show the attention map of Edgeformers. In brief, we find that in higher layers (e.g., Layers 10-11), the attention weights of virtual node tokens are significantly larger than those of original node tokens. As mentioned by the reviewer, the ratio of network tokens to text tokens is $2:l$ in equation (4), where $l$ is the text sequence length. However, the self-attention mechanism can automatically learn to balance the two types of information by assigning higher weights to the corresponding virtual node tokens, so a larger number of tokens representing textual information will not cause network information to be overwhelmed.

---

> > ### Comment · Reviewer_Sciz · 2022-11-21
> > **Response to authors**
> >
> > Thanks for your response.
> >
> > Having reading the responses and other reviewers' comments, I believe some of my questions are answered. However, as reviewer CnjZ points out and I agreed with, the method is still quite complex and the paper's impact, measured by how the community may take inspiration from it, is not fully convincing. Therefore, I would like to keep the current assessment of 6: marginally above the acceptance threshold.

---

> > > ### Author Response · Authors · 2022-11-22
> > > **Response to Reviewer Sciz**
> > >
> > > Thank you so much for your feedback!
> > >
> > > We highly appreciate your time helping us further improve the paper.

---

### Official Review · Reviewer_x3wZ · 2022-10-25

**Confidence:** 3
**Correctness:** 3
**Technical Novelty And Significance:** 2
**Empirical Novelty And Significance:** 2
**Recommendation:** 6

**Clarity, Quality, Novelty And Reproducibility:**

### Clarity
The method section is hard to understand. More clarification and improvements in Figure 1 are needed.

### Quality
Overall, the paper is well-structured.

### Novelty
Edge representation learning with its neighboring node is not a novel concept. This work, however, focuses on applying such an idea to graphs with textual edge attributes first.

### Reproducibility
 The authors provide the code for their method and experiments and explain the details in the Appendix.

**Strength And Weaknesses:**

### Strengths

- **Important problem and valid approach;** This paper addresses an important problem in edge representation learning where the text (document) is provided as an edge attribute. By reflecting graph structure into edge representations using pre-trained language models (PLM), the proposed method successfully handles the given problem.

### Weaknesses

- **Recent baselines are missing;** Relformer [1] is one of the advanced version of Graphformer, which is specialized for link prediction. Otherwise, EHGNN [2] has a similar approach which represents edges based on adjacent nodes and edges with dual hypergraph transformation. In my opinion, such powerful recent algorithms on edge representation learning should be compared as baselines for the proposed method.
- **Hard to understand;** The suggested approach is challenging to comprehend. In order to comprehend the suggested method completely, I read the method section several times. Particularly, Figure 1 is hard to comprehend because there isn't a subscription for each component. Does the transformer layer of PLM have node information appended to it? If not, is Edgeformer another module that integrates the node information into the text representation after the PLM layers? It would be preferable, in my opinion, to incorporate some notations into Figure 1 or include additional, in-depth figures for each method.

[1] Bi et al., Relational Graph Transformer for Knowledge Graph Representations, WWW 2022

[2] Jo et al., Edge Representation Learning with Hypergraphs, NeurIPS 2021

**Summary Of The Paper:**

In this paper, the authors propose the method Edgeformers, which represents the text-attributed edges in a graph.

They utilize the pre-trained language model (BERT) and the concept of virtual node to represent each text in the edge with regards to its adjacent neighboring edges.

Specifically, in Edgeformer-E, they represent each edge attribute with its text and node embedding from their adjacent nodes using a transformer layer.
In Edgeformer-N, the node embeddings are represented as the aggregation of its adjacent edges’ embeddings from Edgeformer-E.

In experiments, they demonstrate the proposed method outperforms previous baselines in both edge classification and link prediction tasks.

**Summary Of The Review:**

This paper has merit on its own in terms of how it affects learning edge representation on graphs with textual edge attributes.

However, to fully demonstrate the validity of their method, more comparisons against recent edge-aware representation learning methods are required.
Additionally, more details on their approach in section 3 could make the writing clearer.

As a result, I give this paper a weak rejection.

---

> ### Author Response · Authors · 2022-11-14
> **Response to Reviewer x3wZ**
>
> Thank you so much for your thoughtful review!
>
> Regarding your questions:
>
> **1. Recent baselines are missing.** Thanks for pointing out the related papers.
>
> **_EHGNN_**: We have added EHGNN (Jo et al., 2021) into the comparison of edge classification since it is designed for edge representation learning. The Macro-F1 results are shown as follows (also available in Appendix A.3),
>
> | Model        | Movie | Apps  | Crime | Children |
> |--------------|-------|-------|-------|----------|
> | EHGNN        | 49.90 | 48.20 | 44.49 | 40.01    |
> | BERT+EHGNN   | 61.45 | 58.86 | 56.92 | 52.46    |
> | Edgeformer-E | 64.18 | 60.67 | 61.03 | 57.45    |
>
> The Micro-F1 results are shown as follows (also available in Appendix A.3),
>
> | Model        | Movie | Apps  | Crime | Children |
> |--------------|-------|-------|-------|----------|
> | EHGNN        | 64.04 | 63.63 | 52.30 | 50.23    |
> | BERT+EHGNN   | 70.73 | 70.79 | 61.66 | 57.97    |
> | Edgeformer-E | 73.59 | 71.28 | 65.86 | 61.71    |
>
> From the result, we can find that Edgeformer-E consistently outperforms EHGNN and BERT+EHGNN. EHGNN-based methods cannot obtain promising results because of two reasons: 1) edge-edge propagation: EHGNN proposes to transform edge to node and node to edge in the original network, followed by graph convolutions on the new hypernetwork. This results in edge-edge information propagation when conducting edge representation learning. However, edge-edge information propagation has the underlying edge-edge homophily assumption which is not always true in textual-edge networks. For example, when predicting the rate of a review text $e_{ui}$ given by user $u$ to item $i$, it is not straightforward to make the judgment based on reviews for $i$ written by other users (neighbor edges); 2) The integration of text and network signals are loose for BERT+EHGNN, since such architectures process text and graph signals one after the other, and fail to simultaneously model the deep interactions between both types of information. By contrast, our Edgeformer-E is designed following the more reasonable node-edge homophily hypothesis and deeply integrating text & network signals by introducing virtual node tokens in Transformer encoding.
>
> **_Relformer_**: For Relformer (Bi et al., 2022), we have cited it in our revised version. Meanwhile, we feel that it cannot be directly applied to textual-edge networks for two reasons: 1) Textual-edge networks are different from knowledge graphs (for which Relformer is proposed): Relformer specializes in representation learning on KGs where nodes and edges are associated with _short texts_ (i.e., entity names and relation names). However, in textual-edge networks, edges are associated with _long_ and _semantic-rich_ texts (e.g., emails and reviews), and nodes are not guaranteed to contain textual information (e.g., many users are not associated with text information and some products have no descriptions). The “[mask] prediction” paradigm for link prediction in Relformer assumes that every node contains text, which is not always the case in textual-edge networks. In addition, Relformer conducts a triple2seq operation to linearize the knowledge triple in the graph by concatenating the triple tokens, but there is no straightforward way to apply such an operation to those nodes without textual information in textual-edge networks and linearizing long edge texts in textual-edge networks can cause the result sequence over the PLM max sequence length (512 for bert-base-uncased); 2) Relformer and Edgeformers are designed for different link prediction settings: Relformer considers the (h, r, ?) task, while Edgeformers study (h, ?). (h is the query node and r is the query relation.)
>
> We would appreciate additional pointers to related papers and we will be happy to discuss them in the revision.
>
> **2. Hard to understand.** Thanks for the advice and we have added some words to Figure 1 and Section 3 to make them clearer. For both architectures, textual information on the edge is first inputted to a Transformer layer to obtain the contextualized representation and then fed into several Edgeformer layers. Due to space limitations, we only show two Edgeformer layers in Figure 1, while in the experiments we do have eleven. With such an architecture, we can initialize the first Transformer layer and all the eleven Edgeformer layers from a 12-layer PLM such as bert-base-uncased. We are happy to address any further suggestions on presentation and clarity.

---

> > ### Author Response · Authors · 2022-11-18
> > **Kind Reminder**
> >
> > Dear Reviewer x3wZ,
> >
> > We want to sincerely thank you again for your involvement and thoughtful feedback! We hope our response addresses your questions.
> >
> > We would like to gently remind you that today is the end of discussion with author involvement. In light of the performance of the newly added baseline methods, we are kindly asking if you are willing to give a higher assessment to our submission. If you have any more thoughts, we are happy to continue our discussion until the deadline.

---

> > ### Comment · Reviewer_x3wZ · 2022-11-21
> > **Response**
> >
> > Thanks to the authors for the response and I appreciate your additional experimental results based on my suggestions.
> >
> > In conclusion, **I raise my rating from 5 to 6**. Now that I've considered this work's impact on their useful but simple idea for resolving tasks based on textual-edge networks, I think it has some benefits to be accepted. But I also agree with other reviewers' opinions that the novelty and impact of the proposed work are still marginal.
> >
> > I provide more thorough justifications for my judgment below:
> >
> > First of all, my concern about the weakness from missing baselines is completely resolved.
> >
> > - I agree with the authors’ claim that Relformer is not directly compared to the proposed method.
> > - The experimental results against EHGNN also sound reasonable considering that EHGNN is not appropriate for edge classification tasks. The authors' justifications for this are provided in Appendix A.3 and are very useful in understanding why Edgeformer functions so well.
> >
> > However, I think the paper (especially, the method section) is still a bit hard to understand.
> >
> > I agree that my previous review is highly abstract. Therefore, I will give more detailed feedback on this so that authors can improve their paper before the camera-ready.
> >
> > - In Section 3.1, Network-aware Edge Text Encoding with Virtual Node Tokens…, last sentence $h_{e_{ij}} = H^{(L)}_{e_{ij}}$ [CLS].
> >
> >     - This is confusing. I guess the authors want to say that $h_{e_{ij}}$ is the representation of [CLS] token from the sequence of embedding $H^{(L)}_{e_{ij}}$.
> >
> >     - Instead, I would suggest using the indexing term (e.g., $H_{e_{ij}, 0}^{(L)}$) where $0$ is the index of [CLS] token.
> > - In Section 3.1, Representation of Virtual Node Tokens
> >     - As far as I understand, the authors introduce virtual node tokens $z_{v_i}^{(0)} \in \mathbb{R}^{64}$ for all nodes in the graph. If so, in the case of the Amazon-Movie dataset, Edgeformer-E requires an additional **node embedding matrix** $\in \mathbb{R}^{173,896 \times 64}$. Am I right? If not, can you elaborate more details on this in the corresponding section?
> >
> > - Figure 1, (b), especially, local network aggregation
> >     - It's still challenging to comprehend. I recommend that the authors enhance Figure 1, particularly the local network aggregation. Giving background colors (like light grey) on the block that represents the local network aggregation in more detail may be preferable.
> >
> > I know that the authors cannot update their paper in stage 2. Therefore, I hope the authors positively consider addressing my suggestions after the rebuttal period.

---

> > > ### Author Response · Authors · 2022-11-22
> > > **Response to Reviewer x3wZ**
> > >
> > > Thank you so much for your feedback!
> > >
> > > Regarding your comments:
> > >
> > > **1. $h_{e_{ij}} = H^{(L)}_{e{ij}}$ [CLS].** Your understanding is correct. We will make it clearer in the next version.
> > >
> > > **2. Virtual node tokens.** Your understanding is correct. In a textual-edge network where nodes are not associated with textual information, Edgeformer-E requires such an additional learnable node embedding matrix. We will elaborate more details and make it clearer in the next version.
> > >
> > > **3. Local network aggregation.** Thanks for pointing it out. We will enhance Figure 1 in our next version.
> > >
> > > We will seriously update our paper based on your suggestions and try our best to make the illustration as clear as possible. We highly appreciate your time helping us further improve the paper.

---

### Author Response · Authors · 2022-11-15
**General Response**

Dear Reviewers,

We sincerely appreciate your valuable feedback and suggestions. Our work has been revised based on your reviews. We highlighted changes in the manuscript using blue color.

We also want to thank the Reviewers for noting the strengths of our paper, namely:
- The problem addressed in our paper is important and well-motivated. (x3wZ, Sciz)
- Our proposed method is substantial and modern. (x3wZ, Sciz, CnjZ)
- The paper is clearly written. (Sciz, CnjZ)
- The empirical results are consistent, solid and convincing. (Sciz, CnjZ)
- The code is open-sourced for reproducibility. (CnjZ)

We have addressed individual questions of reviewers in separate responses. In the revised version, we incorporated all reviewers' suggestions by adding more clarification of our method, more experimental results and baselines, as well as deeper model analysis. Here we briefly outline the updates to the revised submission for the reference of reviewers.

- [Figure 1] We add more explanations to the caption to further illustrate the design of our models. (x3wZ, CnjZ)
- [Section 5: Related Works] We add more related works which are pointed out by reviewers. (x3wZ)
- [Appendix A.3] Some experimental results and analysis for new baselines including EHGNN, BERT+EHGNN, BERT+MaxSAGE and GraphFormers are added. (x3wZ, Sciz)
- [Appendix A.7] We add a self-attention map study section to show how Edgeformers can learn to assign different weights for network tokens and text tokens. (Sciz)
- [code link] We upload our data processing code and processed dataset. (CnjZ)

In closing, we thank the Reviewers again for their time and valuable feedback. If there are further concerns, please let us know, and we will be happy to address them.

---

### Decision · Program_Chairs · 2023-01-20

**Decision:**

Accept: poster

**Justification For Why Not Higher Score:**

BERT was from 2018, so the experiments could be improved by incorporating more recent backbones.

**Justification For Why Not Lower Score:**

This is a clear acceptance since all reviewers voted acceptance.

**Metareview: Summary, Strengths And Weaknesses:**

In this paper, the authors want to design a more expressive edge model for graph neural networks based on transformer models. They propose Edgeformers to improve edge and node representation learning by modeling texts on edges in a contextualized way. The idea is to inject rich text descriptions / context in the edge representation with a transformer architecture instead of just using a simple relation name. The AC thinks that this approach is simple but reasonable, and the authors have agreed that this paper has created a meaningful design to model edges and nodes. The authors did a great job responding to the reviewers, and one reviewer has increased their score after seeing the authors' response. Overall, we recommend an acceptance of the paper based on reviews.

**Note From Pc:**

if the above contains the word "oral" or "spotlight" please see: "oral" presentation means -> notable-top-5% and "spotlight" means -> notable-top-25%. As stated in our emails, we are disassociating presentation type from AC recommendations

**Summary Of Ac-Reviewer Meeting:**

No AC-reviewer meeting was planned since it is not a borderline paper.